# Aptamer-Based Biosensors for the Analytical Determination of Bisphenol A in Foodstuffs

Marica Erminia Schiano [1,†], Avazbek Abduvakhidov [1,†], Michela Varra [1,*] and Stefania Albrizio [1,2,*]

1   Dipartimento di Farmacia, Università degli Studi Federico II di Napoli, Via D. Montesano, 49,
    80131 Naples, Italy; maricaerminia.schiano@unina.it (M.E.S.); avazbek.abduvakhidov@unina.it (A.A.)
2   Istituto Nazionale di Biomolecole e Biosistemi Consorzio Interuniversitario INBB,
    Viale delle Medaglie d'Oro, 305, 00136 Roman, Italy
*   Correspondence: varra@unina.it (M.V.); salbrizi@unina.it (S.A.);
    Tel.: +39-081-678540 (M.V.); +39-081-678607 (S.A.)
†   These authors contributed equally to this work.

**Featured Application: Surveys on BPA levels in specific groups of foodstuffs; design of aptasensors for BPA analogs.**

**Abstract:** Bisphenol A (BPA) is a synthetic compound utilized to manufacture plastics for Food Contact Materials (FCMs) or resins for the inside of food containers. Since it was recognized as an Endocrine-Disrupting Chemical (EDC), its implications in pathologies, such as cancer, obesity, diabetes, immune system alterations, and developmental and mental disorders, have been widely documented. Diet is considered the main source of exposure for humans to BPA. Consequently, continuous monitoring of the levels of BPA in foods is necessary to assess the risk associated with its consumption in one's diet. So far, many reviews have been published on biosensors and aptamer-based biosensors, but none of them focus on their applications in their analyses of bisphenols in food matrices. With this review, the authors aim to fill this gap and to take a snapshot of the current state-of-the-art research on aptasensors designed to detect BPA in food matrices. Given that a new TDI value has recently been proposed by the EFSA (0.04 ng/kg), the search for new sensitive tools for the quantitative analysis of BPA is more topical and urgent than ever. From this perspective, aptasensors prove to be a good alternative to traditional analytical techniques for determining BPA levels in food.

**Keywords:** bisphenols; Endocrine-Disrupting Chemicals (EDCs); biosensors; aptasensors; food safety

## 1. Introduction

Bisphenol A (BPA, Figure 1) is a synthetic chemical with a long history of use. Since 1950, it has been utilized to produce epoxy resins, and it has also been widely applied in the plastic industry as a monomer to synthesize polycarbonate polymers or as an additive in the synthesis of other kinds of plastics, such as polyvinyl chloride (PVC) [1]. BPA is mainly used in manufacturing Food Contact Materials (FCMs), such as storage containers, plastic water bottles, food packaging, and the inner coatings of food cans [2], but it is also contained in electronics equipment, children's toys, dental sealants, thermal paper, and flame retardants.

As a result of its wide usage, BPA is ubiquitous in the environment; therefore, humans can come into contact with this chemical via air, soil, and water [3]. However, diet is still considered one of its main exposure routes, if not the first one [2,4]. There are two substantial paths of foodstuffs contamination: the migration of residual monomers from the linings of cans or from plastic FCMs, and the transfer from the environment to food through the food chain. The extensive presence of microplastics in the environment has led the scientific community to pay special attention to their potential role in the transfer of

several toxic chemicals, including BPA and its analogs, to the food web and, consequently, to humans [5–9].

**Figure 1.** Chemical structure of BPA and its analogs.

Consumers raised concerns for the first time in 2008, when the media brought to the attention of the public that BPA was present in polycarbonate used to manufacture baby bottles. In the same year, the National Toxicology Program (NTP) published its report concerning the risks to human reproduction and development associated with BPA exposure [10]. The number of studies on the adverse effects of BPA on human health have significantly increased since then. At the moment, the toxicity of BPA has been widely demonstrated due to its ability to act as an Endocrine-Disrupting Chemical (EDC). Several negative consequences on human health resulting from the exposure to BPA have been highlighted prevalently on the male and female reproductive apparatuses [11,12]. Recently, other adverse effects have drawn the attention of many researchers who are studying the potential implications of BPA in pathologies such as cancer, obesity, diabetes, immune system alterations, and developmental and mental disorders [13–15]. Children appear to be the most vulnerable category. Prenatal exposure to BPA seems to be critical for both the development of the nervous system and the increase in metabolic disorders in young people [16,17].

In Europe, since the first risk assessment of BPA in 2006, the European Food Safety Authority (EFSA) has constantly reviewed data on BPA's toxicological effects. Based on the EFSA's opinions and updates, the European Commission has regulated over the years the use of BPA in FCMs to protect consumers' health. Currently, Regulation No 2018/213 forbids BPA migration from varnishes or coatings in FCMs intended for use in the manufacturing or packaging of all kinds of foods for infants and young children [18]. The same regulation updated the Specific Migration Limit (SML) for BPA to 0.05 mg kg$^{-1}$ of food, in line with its use in FCMs as well as in varnishes and coatings. In addition, in 2017 BPA was added to the "Candidate List of substances of very high concern for Authorization" because of its endocrine-disrupting properties, in compliance with article 57 of the European Regulation (EC) No 1907/2006 on the Registration, Evaluation, Authorization and Restriction of Chemicals (REACH) [19,20]. In 2018, the BPA entry was upgraded to consider its adverse effects to the environment as an endocrine disruptor. (https://echa.europa.eu/it/-/seven-new-substances-added-to-the-candidate-list-entry-for-bisphenol-a-updated-to-reflect-its-endocrine-disrupting-properties-for-the-environment; accessed on 15 January 2018) [20].

Due to limitations on the use of BPA, the plastic industry has selected BPA chemical analogs as substitutes for the manufacturing of FCMs. Bisphenol F (BPF), Bisphenol B (BPB) and Bisphenol S (BPS) are the most utilized ones (Figure 1). However, several studies have demonstrated that these compounds are as toxic as BPA; therefore, the safety of "BPA-free" materials is not effectively verified [21–29]. BPS is the only regulated substitute among BPA analogs. Regulation EU 10/2011 has indeed settled at 0.05 mg/kg of food as the

SML for BPS [30]. Furthermore, the European Chemical Agency (ECHA) has begun the evaluation of BPB, BPF, and BPS as EDCs. In December 2021, following a re-evaluation of risks associated with exposure to BPA, the EFSA proposed to significantly lower the Tolerable Daily Intake (TDI) to 0.04 ng/kg bw/day for BPA. This decision was based on the analysis of studies published from 2013 to 2018, particularly those focusing on the adverse effects of BPA on the immune system (https://www.efsa.europa.eu/en/news/bisphenol-efsa-draft-opinion-proposes-lowering-tolerable-daily-intake; accessed on 22 February 2022).

For several years, some of the authors of the present review have been working on monitoring BPA and other bisphenols in different commonly used foods by means of LC–FD and LC–MS techniques [31–34]. Generally, the determination of BPA in food is carried out by conventional analytical procedures consisting of a pre-treatment of samples using different extraction techniques, followed by a quali-quantitative analysis mainly by means of LC or GC chromatography, combined with a fluorescence detector or a mass spectrometer [35,36]. However, these procedures are expensive and time-consuming. Furthermore, due to a recent EFSA decision (https://www.efsa.europa.eu/en/news/bisphenol-efsa-draft-opinion-proposes-lowering-tolerable-daily-intake; accessed on 22 February 2022), the need of more sensitive and user-friendly methods for the quantitative analysis of these compounds in foodstuffs is imperative. In that context, the use of sensor technology, notably biosensors, can provide a suitable solution to this challenge [37]. According to the IUPAC definition, a biosensor is "a device that uses specific biochemical reactions mediated by isolated enzymes, immunosystems, tissues, organelles or whole cells to detect chemical compounds usually by electrical, thermal or optical signals" [38]. Biosensors are mainly composed of a recognition element that binds to the target analyte and of a transducer that transforms the biochemical interaction into a detectable and measurable signal. Among all biosensor types, aptamer-based biosensors, named aptasensors, are generally preferred because of their multiple advantages. The use of an aptamer as a recognition element is advantageous for several reasons: aptamers can be easily synthesized and modified to be adapted to a wide range of chemical derivatizations, and they can be re-used with consequent cost reduction. Moreover, aptasensors show high sensitivity and good reproducibility and are easy to use [39].

In our search for scientific works on BPA aptasensors, we noticed that few reviews discussed the issue of BPA aptasensors [39,40], but none of them focused entirely on the application of this technology to the quantitative analysis of BPA in food. Therefore, the present review aims to overcome this lack with the goal of providing a useful tool to understand current state-of-the-art research and to approach this interesting research topic.

## 2. BPA Aptamers: Sequence and Three-Dimensional Structure

Aptamers are sequence-specific single-stranded DNA or RNA oligonucleotides (12–200 nts) that are able to bind to a selected target with high affinity and selectivity. Many studies have shown that aptamers recognize their specific targets employing precise secondary or tertiary structures (duplex, G-quadruplex, stem-loop, etc.). Aptamers are selected by applying the Systematic Evolution of Ligands by EXponential enrichment (SELEX protocol) [41]. This procedure consists of an iterative selection and amplification process to identify, among a large pool of randomly generated single-stranded DNA/RNA sequences, the DNA/RNA molecules able to selectively bind to the immobilized target. After its development, SELEX protocol has been applied to obtain aptamers for different type of targets, varying from small molecules to proteins or also cells [42]. In analytical chemistry, aptamers have been successful applied as key elements to build up sensors able to capture and quali-quantitatively analyze a specific substance in a certain matrix. A general scheme of the principal steps required to build up an aptasensor is depicted in Figure 2.

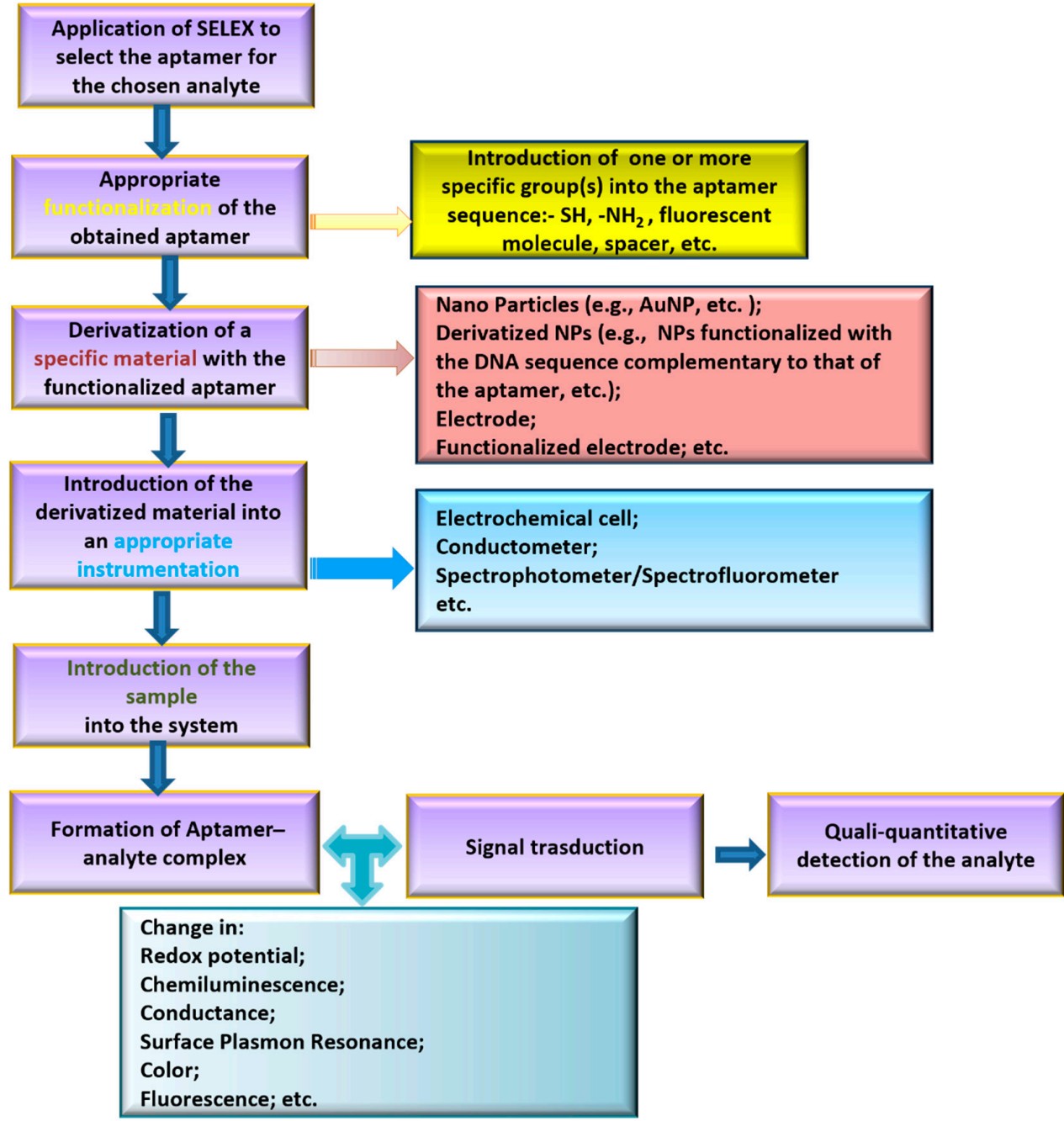

**Figure 2.** Scheme of the general strategy for aptasensor development.

Since the first anti-BPA aptamers were selected by Jo et al. [43], many efforts to develop sensitive BPA aptasensors for their use in different matrices have been made. With its high affinity and selectivity, BPA-Apt-2 (Table 1) is the main adopted sequence in the building up of aptasensors, and despite its 63 nt length, no truncated sequence has been developed until 2017 from the initial aptamer. Starting from the predicted secondary structure of the aptamer, Lee et al. [44] identified that BPA-Apt-3 (Figure 3 and Table 1) was able to bind to BPA with increased affinity and selectivity compared to BPA-Apt-2.

In 2020, a similar truncation protocol was adopted by Jia et al. [45], succeeding in obtaining two differently truncated sequences (Figure 4).

**Table 1.** Aptamers against BPA used in biosensors for BPA detection and quantification in foods.

| Aptamer Binding BPA | | | | | Reference | |
|---|---|---|---|---|---|---|
| Acronym | Length | Sequence | Kd [nM] * | | | |
| BPA-Apt-1 | 60 nts | CCGCCGTTGGTGTGGTGGGCCTAGGGCCGGCGGCGCACAGCTGTTATAGACGCCTCCAGC | not reported | [43] | Table 1, group 2; ID #6 * |
| BPA-Apt-2 | 63 nts | CCGGTGGGTGGTCAGGTGGGATAGCGTTCCGCGTATGGCCCAGCGCATCACGGGTTCGCACCA | 8.3 ** | [43] | Table 1, group 7; ID #3 * |
| BPA-Apt-3 | 24 nts | TTTTTTTTTTGGATAGCGGGTTCC | not reported | [44] | Truncated BPA-Apt-2 |
| BPA-Apt-4 | 38 nts | TGGGTGGTCAGGTGGGATAGCGTTCCGCGTATGGCCCA | 13.17 ± 1.02 *** | [45] | Truncated BPA-Apt-2 |
| BPA-Apt-5 | 12 nts | GGATAGCGTTCC | 27.05 ± 2.08 *** | [45] | Truncated BPA-Apt-2 |
| BPA-Apt-6 | 23 nts | TTTTTTTTTTCCGGTGGGTGGAA | 1190.61 ± 66.05 *** | [44] | Truncated BPA-Apt-2 |

* ID reported by Jo et al. [43] ** Kd value reported in [43]; the value of the same Kd calculated by Ma et al. [46] by means of UV results is 47.3 nM. *** Kd values reported in [45].

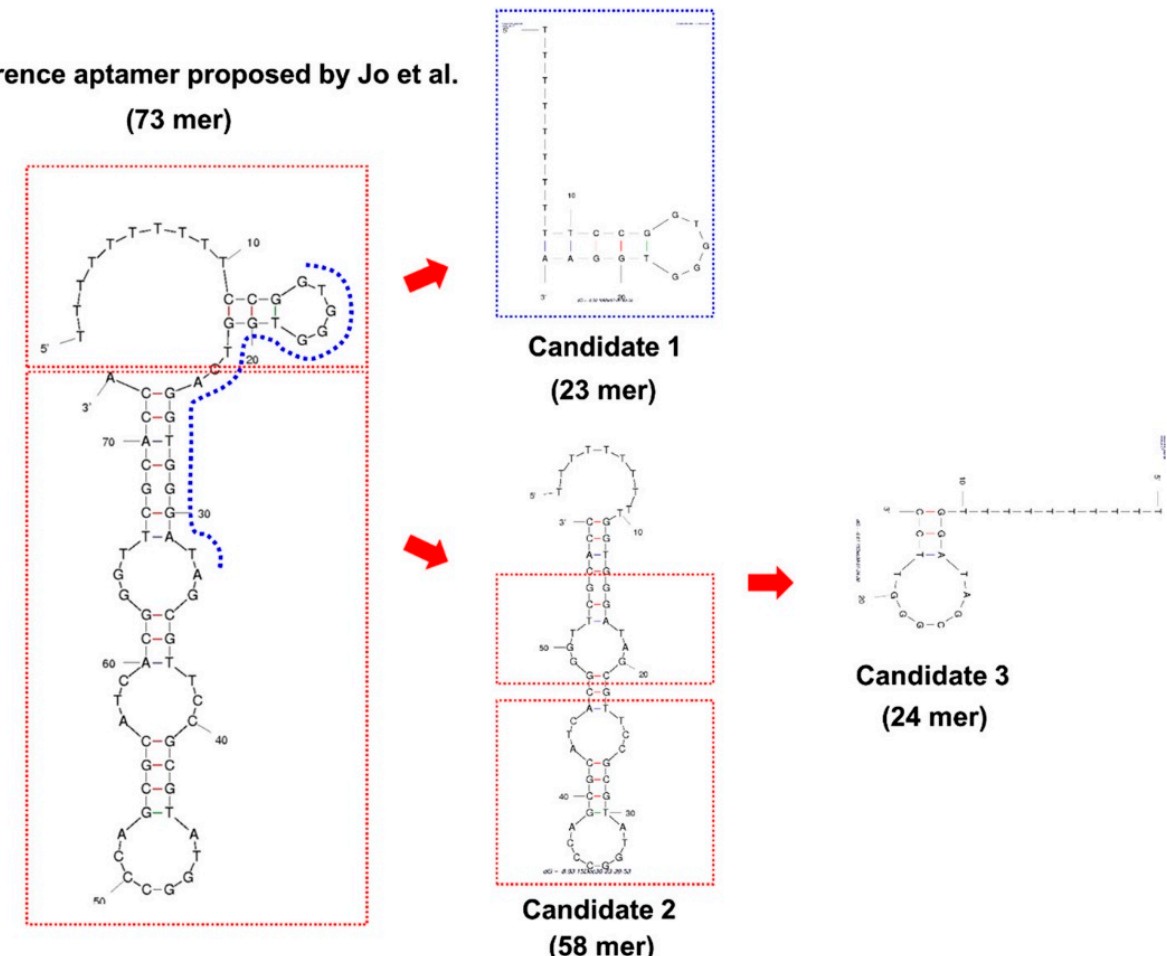

**Figure 3.** The 63 nts BPA-Apta-2 and the corresponding flow chart showing the process used to select the 24 nts truncated aptamer. Ten T residues at 5′-terminus of the initial sequence and of the corresponding truncated ones have been added. [Reprinted with permission from Lee, E.H.; Lim, H.J.; Lee, S.D.; Son, A. Highly Sensitive Detection of Bisphenol A by NanoAptamer Assay with Truncated Aptamer. ACS Appl. Mater. Interfaces 2017, *9*, 14889–14898. Copyright 2017, ACS American Chemical Society].

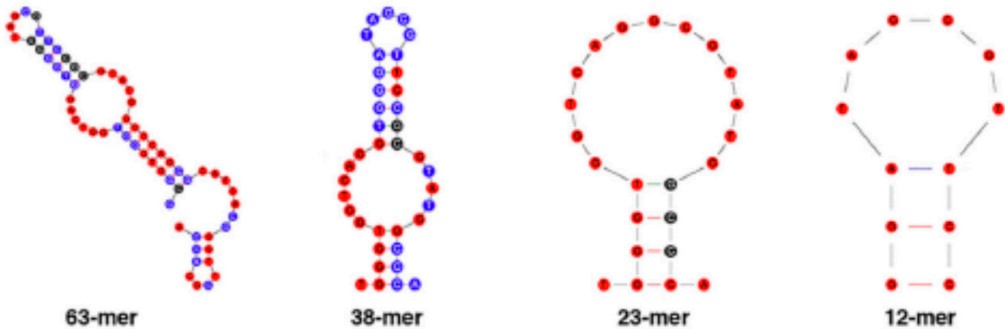

**Figure 4.** The flow chart from BPA-Apt-2 to the two selected 38 and 12 nts truncated aptamers [45]. [Reprinted from Food Chem, 317, Jia, M.; Sha, J.; Li, Z.; Wang, W.; Zhang, H. High affinity truncated aptamers for ultra-sensitive colorimetric detection of bisphenol A with label-free aptasensor 126459–12467. Copyright 2020, with permission from Elsevier].

The comparison between the three-dimensional structure predicted for BPA-Apt-3 and that of BPA-Apt-5 shows the presence of the same stem, formed by the 5′-GGA and 3′-TCC, and a differently long loop containing a common AGCGT sequence in both the folded aptamers. These two works confirm that SELEX alone is unsatisfactory for obtaining an efficient and cost-effective aptamer. Therefore, further studies are necessary to identify the exact binding mode of BPA to its aptamers and/or to the shortened aptamer derivatives as well as the secondary structures adopted in solution by the aptamer(s) alone and in the BPA–aptamer complex(es). Indeed, only a few studies reported the molecular modeling of aptamers and their BPA complexes [44,45], whereas in most cases, the stem-loop aptamer structures have been derived based on predictive DNA programs, and only in few cases they were confirmed by circular dichroism analysis. Notably, some authors mentioned the ability of BPA to induce the folding of BPA-Apt-2 (Table 1) into G-quadruplex(s) [47–50], although this hypothesis requires further experimental evidence.

### 3. Aptamer-Based Biosensor Applications Tested for the Determination of BPA in Real Food Samples

So far, aptamer-based biosensors have not been applied to an extensive monitoring of the levels of BPA in food samples. However, several authors reported the designs of various kinds of aptasensors to be specifically used in the detection of BPA in food samples. After their construction, all the aptasensors were tested to verify their feasibility in the analysis of real samples. All foods used in the applicability tests were liquids, mostly tap or mineralized water [45,49,51–61], followed by milk and fruit juices [47,53,59,60,62,63] and red wine samples [64]. Either no treatment (water samples) or just a quick pre-treatment of samples was necessary. Fruit juices were filtered to remove any residues of solid particles. Milk samples required a slightly time-consuming pre-treatment to eliminate fats by centrifugation and to eliminate proteins by precipitation and further centrifugation. Mirzajani et al. [65] analyzed two different brands of canned food (green beans and sweet peas), but they carried out the experiments only on the liquid portion of the cans. However, studies in the literature demonstrated that, when canned foods are analyzed, the BPA concentrations are higher in the solid portion than those detected in the liquid portion [33,66]. Especially in the case of canned seafoods, this should be considered in relation to the high probability of contamination due to the ingestion of microplastics by fish in the marine environment [6,8,9]. At last, Lee et al. [67] investigated the application of their colorimetric aptasensor directly on grains of commercially available steamed rice pre-treated with a solution of BPA.

The limit of detection (LOD) and recovery at different concentrations of BPA were the main performance parameters that the authors evaluated. In addition, the selectivity of each aptasensor was generally assessed vs. possible interferent substances or BPA analogs, including some compounds currently used as substitutes of BPA in plastic manufacturing. In several studies the reproducibility was also measured. Both selectivity and reproducibility proved to be good.

In Table 2 the LOD and recovery percentage values obtained in the analysis of real samples are reported, along with a brief description of the developed aptasensor strategies (see next paragraph). The LODs, estimated mostly using BPA solutions at different concentrations, were generally very low. Jia et al. [45] compared the performance of their aptasensor to other more traditional methods. They showed that the LOD obtained by the aptasensor was generally lower, except for the case of GC–MS and electrochemical methods. Nonetheless, the authors concluded that their detection technique was preferable considering the undoubted advantages offered by the aptasensor, such as a low cost and a simple detection process in addition to a still-high sensitivity.

Generally, aptasensors tested in food samples showed good stability. Only Ye et al. found that their biosensor was not very stable when compared to the LC–MS/MS method, although they suggested that this problem could be overcome by carefully checking the temperature and buffering solution [59].

## 4. Detection Techniques Applied in the Analysis of Real Food Samples

### 4.1. Electrochemical Aptasensors

An electrochemical sensor measures an electrical parameter (potential, current, impedance, etc.) whose value can be related to the target analyte concentration in solution. Generally, it consists of a three-electrode electrochemical cell in which the working electrode (WE) constitutes the key element for analyte detection (Figure 5a). WE can be "directly" or "indirectly" functionalized with an aptamer that is able to selectively recognize a specific analyte, resulting in an electrochemical aptasensor. Direct functionalization involves the aptamer being covalently linked to a specific material that covers the WE, whereas in the indirect WE functionalization, the aptamer hybridizes with its complementary sequence, which, in turn, is somehow linked to the WE. Regardless, the formation of an aptamer–analyte complex lacks any detectable electrochemical signal, and, in Faradaic electrochemical aptasensors, a redox probe (methylene blue, ferrocene, horse radish peroxidase, etc.) needs to be employed. In accordance with the adopted sensing strategy, the redox probe can be conjugated to the aptamer or included in a specific step of the WE functionalization process. Alternatively, the redox probe can be externally used, adding it into the test solution ($[Fe(CN)6]^{3-/4-}$, $Ag^{+/0}$, etc.). Different working electrodes for BPA recognition were prepared and used in a conventional three electrode cell in which a Ag/AgCl (in a saturated KCl solution) and a Pt wire are generally used as reference and counter electrodes, respectively.

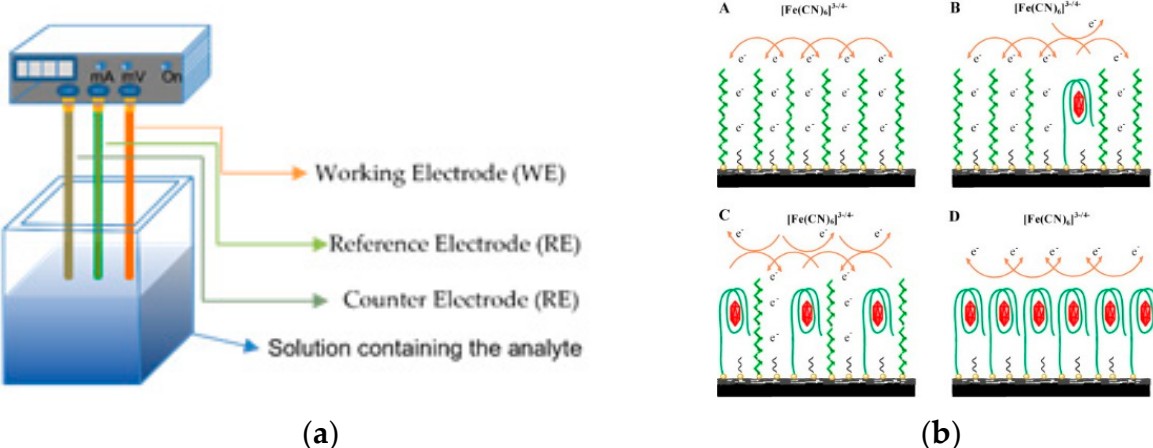

(**a**)　　　　　　　　　　　　　　　　　(**b**)

**Figure 5.** (**a**). Schematic representation of a three-electrode cell. The difference in potential is measured between the working electrode (WE, orange bar) and the reference electrode (RE, light green). The current is equal in absolute value but of opposite in sign between the WE and the counter electrode (CE, green). Compared to a two-electrode cell, in a three-electrode cell the current and the potential are decoupled, led to a more accurate measurement of these cell parameters. (**b**). The sensing mechanism using the Zhou et al. WE. In absence of BPA, the aptamers are unfolded and the exchange of electrons between the WE and the ferrycianide probe is allowed. After the formation of BPA-aptamer complexes the exchange of the electrons with the probe was umpaired, resulting in a decrease of the peak current. [Reprinted from J. Food Chem., 162, Zhou, L.; Wang, J.; Li, D.; Li, Y. An electrochemical aptasensor based on gold nanoparticles dotted graphene modified glassy carbon electrode for label-free detection of bisphenol A in milk samples. 34–40. Copyright 2014, with permission from Elsevier].

Zhou et al. [47] coated a glassy carbon electrode (GCE) with graphene oxide (GO) and gold nano particles (AuNPs). (Figure 5b). Deiminiat et al. [53] built an electrochemical aptasensor by functionalizing a gold electrode with multiwall carbon nanotubes/gold nanoparticles (f-MWCNTs/AuNPs) to realize a nanocomposite film on which the aptamer was immobilized. Ma et al. [46] built a very sensitive electrochemical aptasensor by coating a boron-doped diamond (BDD) layer with AuNPs and by modifying the latter with

aptamers. Tsekeli et al. [50] developed a GCE coated with silver nanoparticles/carbon nanofiber composite (AgCNFs) and functionalized it with an aptamer via a disulfide bond between the aptamer and silver nanoparticles.

In all these cases the anti-BPA aptamers covalently linked on the WE produced tunnels on the surface of the electrode that acted as gates for the redox probe ($[Fe(CN)_6]^{3-}/^{4-}$). In the absence of BPA, the gates were opened, and the passage of electrons to the electrode surface was allowed. Upon the addition of BPA, the conformation of aptamer changed, and the gates became closed. The electron flow was hampered by both physical hindrance to the passage of the redox probe and by a repulsion charge with the negatively charged backbone of the aptamer–BPA complexes. An increment of BPA concentration increased the formation of closed aptamer gates and subsequently reduced the number of open tunnels, decreasing the electrochemical signal (Figure 5b, panels BC). All these strategies are simple and cost-effective. However, the use of an external probe presents the same limitations, especially for repetitive detections, due to possible electrodeposition of contaminants on the electrode. Nonetheless, the necessity to add reagents during the analysis could further increase the variability of results.

A singular and simple DNA- and enzyme-based BPA electrochemical aptasensor was developed by Abnous et al. [68]. A commercial screen-printed gold electrode (SPGE) was first functionalized with both the BPA aptamer and its complementary strand, with the latter extended at the 3′-end with a polyA tract. The use of the terminal deoxynucleotidyl transferase (TdT) in the presence of dT triphosphate led to the addition of T residues at the 3′-end of the aptamer coated on the electrode. Then, polyT at 3′-end of the aptamer and the polyA at the 3′-end of the complementary sequence hybridized, forming a bridge on the surface of the electrode. This negatively charged physical barrier hampered the access of redox mediators to the electrode surface, producing a weak current signal. On the other hand, in the presence of BPA, the aptamers were sequestered. The flow of events which led to the formation of bridged duplexes was disrupted, and an increase in the current signal occurred. Consequently, with respect to the cases described above, the measured electrochemical parameter increases after the binding of BPA to the immobilized aptamer. However, aside from the external redox probe, other reagents (TdT and dT) should be added in the system that may further contribute to the variability of the results in repetitive detections.

Very sensitive electrochemical aptasensors, which also include the use of external redox probes, have been recently built up by Ensafi et al. [69] and by Farahbakhsh et al. [60]. In the first case, an increase in sensitivity has been obtained by immobilizing a highly conductive polipyrrole (PPY), which was immobilized on a GCE. Firstly, the surface of the GCE was covered with AuNPs. Then, a BPA aptamer was immobilized on the electrode surface in the presence of BPA. Finally, the pyrrole was electropolymerized on the surface of the resulting electrode. After the removal of BPA, the functionalized GCE was used for analytical determinations. The latter case relied upon the use of aptamer-modified magnetic $Fe_3O_4$/Au nanoparticles to obtain signal amplification and a catalytic effect on the reduction of $Ag^+$ to $Ag^0$. A functionalized aptamer–gold electrode was used to capture the BPA in solution. The aptame-modified magnetic $Fe_3O_4$/Au nanoparticles solution was then dropped onto the electrode, and the BPA molecules on the gold-electrode could attach to many aptamers on $Fe_3O_4$/Au/Apt nanoparticles. As a result of the increment of its concentration in the tested solution, the BPA amount captured on the aptamer-modified electrode rose, increasing the sequestered aptamer-modified $Fe_3O_4$/Au nanoparticles. The $Fe_3O_4$/Au nanoparticles were detached from the electrode and transferred to a solution containing $Ag^+$ ions. As a result, they catalyzed the reduction of $Ag^+$ and then captured $Ag^0$. Finally, the electrode was used in a three-electrode cell. A positive-going differential pulse potential scan was applied to oxidate the $Ag^0$ to the AgCl, and the magnitude of the stripping anodic signal of the $Ag^0$ was related to the concentration of the BPA. A very low LOD level resulted in BPA detection by both aptasensors; however, the first case implies

a reduced number of steps during the analyses, which may correlate with cost-effective aptasensor production and limiting data variability in repetitive BPA measurements.

Beiranvand et al. [58] used a covalently functionalized acriflavine (ACF) redox probe starting from a GCE that was firstly coated with carbon nanotubes-COOH (CNTs-COOH) and then decorated with golden-platinum nanoparticles (Au-PtNPs) and finally derivatized with ACF. A phosphoramidate bond between ACF-NH$_2$ and the phosphate group at the 5′-end of the aptamer led to the final electrode functionalization. The authors demonstrated that each sequential modification of the electrode, GCE→GC/CNTs-COOH→GCE/CNTs/Au-PtNPs→GC/CNTs/Au-PtNPs/ACF, gradually and sensitively increased the cyclic voltammetry (CV) signal up to the coating of the aptamer on the GC/CNTs/Au-PtNPs/ACF electrode, which, conversely, negatively affected the signal. The aptamer–BPA complex formation triggered further decreasing of the CV signal. Despite this strategy implying the use of chemical reactions to conjugate the redox probe to the aptamer, the use of an amide bond can guarantee good stability of the resulting functionalized WE in repetitive detections. Li et al. [70] used the methylene blue dye as an aptamer intercalating redox probe to avoid aptamer chemical conjugation, a time-consuming and cost-increasing procedure. Indeed, the working electrode was built on a PET film coated with Au and wrapped with carbon tape to achieve an adhesive surface that captured multiwalled CNs in non-covalent way. The obtained electrode surface was then coated with Au and functionalized with the selected aptamer via a disulfide bond. The redox-active dye methylene blue was then intercalated into the aptamer, and the final working electrode was ready to be used. In the absence of BPA, the intercalated dye resulted in a strong electrochemical signal. Once BPA had bonded to its aptamer, the intercalated dye was released, and the electrochemical signal decreased. Both the use of multi-walled CNs and an aptamer–dye complex contributed to a significant signal amplification, allowing for very sensitive BPA detection. However, the use of methylene blue as the intercalating reagent instead of the conjugated reagent, even though simply the chemical functionalization of the WE, also introduces new steps during the analysis, namely the incubation time with the dye to obtain the intercalated methylene blue/aptamer complex, and the subsequent washing of WE to remove the excess of dye molecules, which may contribute to increasing the variability of the results of repetitive measurements.

Liu et al. [55] and Shi et al. [64] used enzymes to detect or to amplify, respectively, BPA detection. In the first case, the authors used the system horseradish peroxidase (HRP)/hydroquinone (HQ)/H$_2$O$_2$ to generate the electrochemical signal associated with the formation of the aptamer–BPA complex on the WE. In particular, they functionalized a GCE with gold nanoparticles (AuNPs) carrying a thiolated BPA aptamer. A biotin-modified complementary DNA probe (cDNA) was used to form double-stranded DNA (dsDNA) on a GC/AuNPs/Apt electrode. A subsequent interaction between the biotin and the avidin on streptavidin-modified horseradish peroxidase gold nanoparticles (avidin-HRP-AuNPs) generated a signal of hydroquinone (HQ) oxidation in the presence of H$_2$O$_2$. The oxidized HQ turned back into HQ itself, exchanging electrons with the electrode and causing the amplification of the reduction current. Upon the highly specific interaction between the BPA and its aptamer, cDNA was released from the electrode surface, and BPA was immobilized on the sensing interface. The captured amounts of avidin-HRP-AUNP diminished with the increase in the BPA concentration and produced a series of decreasing catalytic peak currents (Figure 6). In the latter case, Shi et al. combined a DNA-hybridization strategy and enzymes to obtain a signal amplification. Aptamer strands were firstly attached on magnetic beads (MBs) using the amide-coupling method and then were hybridized with complementary DNA (message DNA, mDNA). The presence of BPA prompted the release of mDNA strands from the magnetic beads. The dissociated mDNA strands were hybridized with capture DNA (cDNA) immobilized on a gold nano-porous microelectrode (np-Au), and the cleavage reaction was activated with Exo III. cDNA strands were subsequently digested, and

the dissociated mDNA was used for a new cycle. The amount of cDNA strands on the microelectrode decreased with the increase in the target concentration. Finally, with the assistance of help DNA (hDNA), the residual cDNA strands on the np-Au surface were recognized with report DNA/gold nanoparticles (rDNA/AuNPs), containing methylene blue, to generate an electrochemical signal for quantitative target detection. The adopted three-electrode system consisted of Au wire as the working electrode, Zn plate as the auxiliary electrode, and Zn wire as the reference electrode. This latter approach is clever, but nonetheless, it is accomplished by the employment of several different materials which, also considering the obtained LOD (Table 2), may significantly increase the costs-benefits ratio in a large-scale aptasensor application.

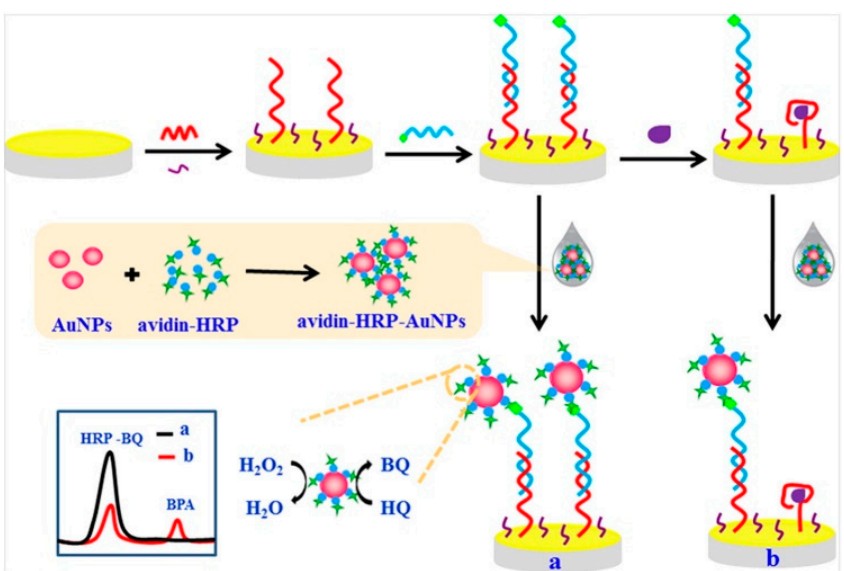

**Figure 6.** The sensing mechanism developed by Liu et al. [55]. The GCE/AuNPs/Apt WE reacted with complementary DNA conjugated with biotin to form a dsDNA on the surface of WE. AuNP/HRP/Avidin was bound through biotin/avidin recognition. Horseradish-peroxidase (HRP) catalyses the hydroquinone HQ) oxidation (BQ), in presence of $H_2O_2$. The oxidate hydroquinone comes back to HQ recovering electrons from the electrode, thus causing the signal amplification. In presence of BPA, Aptamer-BPA complex is formed, and the complementary DNA is released, thus decreasing the catalytic peak current. [Reprinted from J. Electroanal. Chem., 781, Liu, Y.; Liu, Y.; Liu, B. A dual-signaling strategy for ultrasensitive detection of bisphenol A by aptamer-based electrochemical biosensor, 265–271, Copyright 2016, with permission from Elsevier].

*4.2. Electrochemiluminescence-Based Aptasensor*

Electrochemiluminescence (ECL) is a light emission process that occurs in a redox reaction of electrogenerated reactants. This detection technique shows several advantages over other methods, including a high signal-to-noise ratio and measurements with minimal or no background signal. Consequently, different ECL-sensing approaches have been recently employed to develop new, simple, low cost, and portable biosensors, including aptasensors [71,72]. In this frame, Guo et al. [54] developed a chemiluminescent nanomaterial NaYF$_4$:Yb,Er/Mn UCNPs and used it in an ECL-based aptasensors. In this strategy a thiolated BPA aptamer was covalently immobilized on a gold nanoparticle (AuNPs)-modified electrode and paired with complementary DNA labeled with NaYF4:Yb,Er/Mn UCNPs. The formation of the aptamer/cDNA-NaYF4:Yb,Er/Mn UCNPs double helix on the AuNPs/apt electrode generated a strong and stable chemiluminescent signal. The binding of BPA to the coated aptamer allowed for the release of conjugated NaYF4:Yb,Er/Mn UCNPs-DNA, and consequently the electrochemiluminescent signal decreased.

Ye et al. [59] used [Ru(phen)$_3$]$^{2+}$ as the indicator of the ECL signal. The aptamer coated on a gold electrode was hybridized with the complementary DNA to form dsDNA.

[Ru(phen)$_3$]$^{2+}$ intercalated into the grooves of dsDNA; therefore, high ECL intensity was detected from the electrode surface. The binding of BPA to the aptamer disrupted the dsDNA, and [Ru(phen)$_3$]$^{2+}$ was released from the electrode surface, leading to a decrease in the ECL signal.

Zhang et al. [73] used hybridization chain reaction (HCR) technology to realize the binding site of the electrochemiluminescent [Ru(phen)$_3$]$^{2+}$ on the indium-tin-oxide (ITO) working electrode (Figure 7). In particular, cDNA was coated on the surface of the electrode and then hybridized with BPA aptamer to form dsDNA. In the presence of BPA, the aptamer was released, and the free cDNA triggered the formation of long dsDNA by HCR. Therefore, [Ru(phen)$_3$]$^{2+}$ could intercalate the long dsDNA, causing an enhancement of the electrochemical signal.

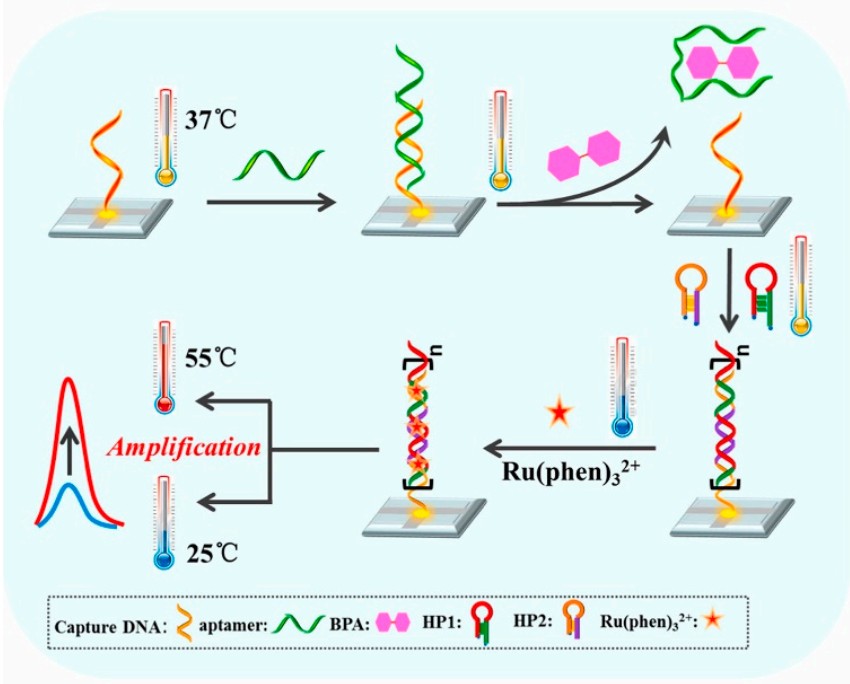

**Figure 7.** Schematic representation of the strategy used by Zhang et al. [73] to build up the electrochemiluminescent aptasensor through the functionalization of the indium-tin-oxide (ITO) working electrode. [Reprinted from Biosensors and Bioelectronics, 129, Zhang, H.; Luo, F.; Wang, P.; Guo, L.; Qiu, B.; Lin, Z. Signal-on electrochemiluminescence aptasensor for bisphenol A based on hybridization chain reaction and electrically heated electrode, 36–41, Copyright 2019, with permission from Elsevier].

Very low LOD values have been obtained in the last two cases (Table 2). However, it should be carefully considered that these chemiluminescent materials are associated with the use of heavy and toxic metals. In their use on a broader scale, they have the disadvantage of producing polluting waste.

### 4.3. Capacitive Aptasensors

Capacitive biosensors can be classified as non-Faradaic electrochemical biosensors because the use of a redox probe is avoided in this system [74]. The sensing event consists of a change of dielectric properties and/or thickness of the dielectric layer at the electrolyte–electrode interface, in which an analyte interacts with its receptor immobilized on the insulating dielectric layer. The sensing event can be monitored by measuring the interfacial capacitance. Mirzajani et al. [65] obtained a BPA capacitive aptasensor by means of printed circuit board material. The BPA level in solution was evaluated by measuring the sample/electrode interfacial capacitance of the sensor. The alternating current electrothermal effect (ACET) was generated during capacitance measurements by a plate WE, and

it significantly contributed to an increase in the rate of BPA transport towards the electrode surface, thus reducing the time of the analysis. The final BPA aptasensor (Figure 8) exhibits different advantages compared to Faradaic electrochemical biosensors developed for BPA detection in food matrices: (i) the adopted strategy to functionalize the plate electrode is simple and low-cost; (ii) the use of an external redox probe is avoided; (iii) the incubation time of the functionalized WE with BPA-containing solution is shortened by electrothermal effects during capacitance measurements; and (iv) repetitive analyses can be rapidly performed. However, as the sensing mechanism is conceived in capacitance biosensors, even slight changes in the environment surrounding the electrodes can significantly affect the measurements. Thus, the experimental condition of BPA detection should be strictly controlled. This phenomenon can limit the applicability of the system in more complex food matrices than in those tested (solid portion of canned food).

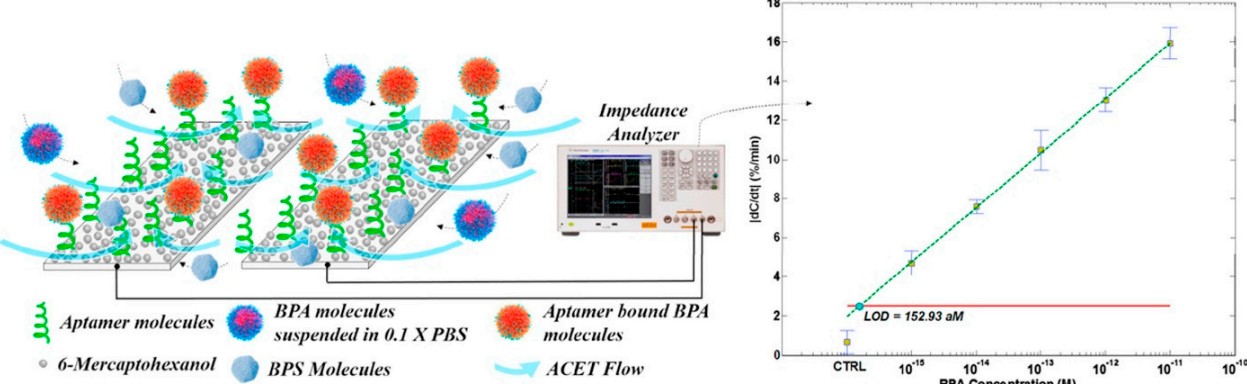

**Figure 8.** A schematic representation of the capacitive aptasensor by Mirzajani et al. [65]. [Reprinted from Biosensors and Bioelectronics, 89, Mirzajani, H.; Cheng, C. A highly sensitive and specific capacitive aptasensor for rapid and label-free trace analysis of Bisphenol A (BPA) in canned foods 1059–1067, Copyright 2017, with permission from Elsevier].

### 4.4. Optical Aptasensors

In these devices, the binding of BPA to its specific aptamer is transduced in an optical signal consisting of a change of a specific light parameter, such as intensity, phase, color, or polarization. Chung et al. [51] produced a surface-enhanced Raman-scattering (SERS) aptasensor functionalizing AuNPs with thiolate probe DNAs and then coating the resulting DNA-AuNPs with a silver shell to enhance the Raman signal. Finally, the 18 nt sequence (CACCTGACCACCCACCGG) of the probe DNAs on Au/AgNPs was hybridized with Cy3-labeled aptamers. The closeness of Cy3 at the 5′-end of the aptamer to the silver shell enhanced the SERS signal. Cy3-labeled aptamers were released from the nanoprobe surface after their binding to BPA, causing a decrease in the SERS signal. Another SERS-based aptasensor was developed by Feng et al. [56]. They used a gold nanoparticle-nanorod hetero-assembled material to obtain amplifications of the Raman scattering signals. In this strategy, gold nanoparticles were functionalized with the aptamer, whereas the nanorods were functionalized with the complementary DNA. The two types of nanomaterials were then assembled via the formation of dsDNA, resulting in a new material that produced a high Raman scattering signal. In the presence of BPA, the material disassembled, producing a decrease in the Raman signal.

Ren et al. [62] built up a sensing device composed of a Liquid Crystal (LC) coated with a BPA aptamer and then used polarized light microscopy to monitor the changes in the LC structure after the binding of BPA to the aptamer.

Li et al. [75] used a detection strategy based on the inner filter effect (IFE) on Cadmium Tellured Quantum Dots (CdTe QDs) caused by non-conjugated AuNPs. BPA aptamers could bind to AuNPs without changing the IFE of AuNPs on CdTe QDs,

whereas CdTe QDs fluorescence was restored after the binding of BPA to the aptamer coating on the AuNPs.

Recently, a series of different, simple, and efficient colorimetric methods for BPA-quantification in foodstuffs has also been developed. Zhang et al. [76] exploited the ability of the cationic polymer poly(diallyldimethylammonium chloride) (PDDA) to induce the aggregation of AuNPs when added to a solution containing dispersed AuNPs. In the absence of BPA, the addition of an appropriate quantity of PDDA to a solution of dispersed AuNPs and BPA aptamers left the color solution unchanged, because of the electrostatic interactions between PDDA and the aptamer. In the presence of BPA, the aptamer was sequestered, and the released PDDA caused AuNP aggregation. The switch between free and aggregate forms of AuNPs caused a BPA-concentration-dependent color change in the solution. Similar colorimetric methods were developed by Jia et al. [45] and by Lee et al. [67]. They conjugated differently truncated BPA aptamers to AuNPs, which underwent electrolyte-induced aggregation in the presence of very low levels of BPA.

Liu et al. [77] described a direct aptamer Fluorescence Anisotropy method for BPA detection by using a single tetramethylrhodamine (TMR)-labeled 35 nt aptamer probe (5′-CCGCCGTTGGTGTGGTGGGCC-T-(TMR)AGGGCCGGCGG-3′). The binding of BPA to the aptamer generated a BPA-concentration-dependent decrease in the fluorescence signal. More recently, Pan et al. [78] developed a new fluorescence-based aptasensor combining a dual-labeled DNA substrate and a DNA circuit for signal amplification. A catalytic $Mg^{2+}$-dependent DNAzyme was formed through a synergistical DNA hybridization strategy. The process started with the binding of BPA to the aptamer that produced the release of a DNA segment from a DNA/aptamer double helix. The released DNA segment became the essential part of a series of double strand hybridization processes from which the DNAzyme was produced. The obtained DNAzyme cleaved the dual-labeled substrate DNA into two segments and induced the separation of the fluorophore and quencher, generating a fluorescence response whose intensity depended on the BPA level.

Berberine has also been used as a fluorescence signal messenger [79]. In a buffered solution, berberine resulted in a weak fluorescence signal that increased with the addition of BPA aptamers and decreased again in the presence of BPA.

Of all optical aptasensors investigated for the detection of BPA in food, these appear to be easy to use, and in some cases they show a very low LOD value (Table 2) [45]. Consequently, they seem to be particularly promising for applications in routine analyses.

### 4.5. Other Detection Techniques

In 2018, Zhu et al. [56] developed a real-time quantitative polymerase chain reaction (RT-qPCR)-based biosensor. In this device, a biotin-modified aptamer DNA was fixed on the inner wall of a streptavidin-coated PCR tube via biotin–avidin interactions, and the DNA template was hybridized with the aptamer DNA. Upon the addition of BPA, the DNA template was released and affected the PCR process depending on the BPA concentration.

Qiao et al. [49] developed a photo-electrochemical (PEC) sensor based on an Indium Tin Oxide (ITO) electrode coated with Au/ZnO pencils and functionalized with an aptamer. The PEC process converted photons in electricity by charge separation, and subsequent charge transfers were activated by light absorption. Due to the presence of Au in the ZnO nanopencil modified ITO electrode, the photo-stimulated current sensitively increased respect to that which was obtained using the ZnO/ITO electrode alone. After coating the BPA aptamer on the Au/ZnO nanopencil, the decrease in the intensity of the PEC signal generated on the aptamer/Au/ZnO/ITO electrode depended on the increase in BPA concentration.

**Table 2.** Summary of the aptasensor applications tested on real food samples.

| Type of Aptasensor | Probe | | Detection Technique | Aptamer Sequence | Food Tested for Practical Application | Recovery (%) | LOD (M) | References |
|---|---|---|---|---|---|---|---|---|
| | Method of Use | Type | | | | | | |
| Electrochemical | External | $[Fe(CN)_6]^{3-/4-}$ | Differential Pulse Voltammetry | BPA-Apt-2 | liquid milk | 96–116 | $5.0 \times 10^{-9}$ | [47] |
| | | | | | milk powder | 90–112 | | |
| | | | Square Wave Voltammetry | BPA-Apt-2 | mineral water | 98.00–102.00 | $5.0 \times 10^{-11}$ | [53] |
| | | | | | milk | 96–103 | | |
| | | | | | orange juice | 96–106 | | |
| | | | Impedance Spectroscopy | BPA-Apt-2 | milk | 92–108 | $7.2 \times 10^{-15}$ | [46] |
| | | | Square Wave Voltammetry | BPA-Apt-2 | water | 102.6–123.6 | $3.9 \times 10^{-10}$ | [50] |
| | | | Differential Pulse Voltammetry | BPA-Apt-2 | tap water | 88.6–97.3 | $1.5 \times 10^{-11}$ | [69] |
| | | | | | grape juice | 89.5–95.8 | | |
| | | | Impedance Spectroscopy | BPA-Apt-2 | tap water | 94 | $8.0 \times 10^{-17}$ | [69] |
| | | | | | milk | 96.0–102.0 | | |
| | | $Ag^{0/+}$ | Stripping Voltammetry | BPA-Apt-2 | tap water | 95.2–108.4 | $6.0 \times 10^{-16}$ | [60] |
| | | | | | mineral water | 98.0–103.6 | | |
| | | | | | milk | 96.0–103.8 | | |
| | | | | | orange juice | 105.4–106.4 | | |
| | Conjugated | Acryflavine | Differential Pulse Voltammetry | BPA-Apt-2 | water | 94–103.6 | $3.5 \times 10^{-14}$ | [58] |
| | Intercalated | Methylene blue | Square Wave Voltammetry | BPA-Apt-2 | drinking and tap water | 99.1–103.3 | $8 \times 10^{-15}$ | [70] |
| | | | | | milk | 103.3–106.2 | | |
| | Biotine–avidine interaction | $SHP/HQ/H_2O_2$ | Differential Pulse Voltammetry | BPA-Apt-2 | tap water | 95–105 | $4.1 \times 10^{-13}$ | [55] |
| | Intarcalated | Methylene blue | Differential Pulse Voltammetry | BPA-Apt-2 | red wine | 95.0–103.0 | $4.4*10^{-11}$ | [64] |
| | - | - | Capacitance | BPA-Apt-2 | canned food | 75.1–106.1 | $1.53 \times 10^{-16}$ | [65] |

Table 2. *Cont.*

| Type of Aptasensor | Probe | | Detection Technique | Aptamer Sequence | Food Tested for Practical Application | Recovery (%) | LOD (M) | References |
| | Method of Use | Type | | | | | | |
|---|---|---|---|---|---|---|---|---|
| Electrochemical luminescence | Conjugated | NaYF$_4$:Yb,Er/Mn UCNPs | Chemiluminescence | BPA-Apt-2 | mineral water | 0–102.50 | $1.6 \times 10^{-10}$ | [54] |
| | Intercalated | [Ru(phen)$_3$]$^{2+}$ | Chemiluminescence | BPA-Apt-6 | mineral water | 96–105 | $7.6 \times 10^{-14}$ | [59] |
| | | | | | milk | | | |
| | | | | | canned juices | | | |
| | Intercalated | [Ru(phen)$_4$]$^{2+}$ | Chemiluminescence | BPA-Apt-2 | milk | 98.4–105 | $1.5 \times 10^{-12}$ | [73] |
| | | | | | orange juice | 96.0–101.2 | | |
| | | | | | coconut juice | 97.3–103 | | |
| Photo-electrochemical | Covered on the ITO electrode | Au/ZnO | Photo-induced current | BPA-Apt-2 | drinking water | 97.3–108.4 | $5.0 \times 10^{-10}$ | [49] |
| | | | | | milk | 96.2–105.7 | | |
| Optical | Conjugated | Cyanine dye (Cy3) | SERS | BPA-Apt-2 | tap water | not reported | $1.0 \times 10^{-14}$ | [51] |
| | Hybridized | hetero-assembled material (nanoroad-nanoparticles) | SERS | BPA-Apt-2 | tap water | 91–95.3 | $1.64 \times 10^{-11}$ | [56] |
| | Conjugated | carboxyfluorescein | FRETe | BPA-Apt-2 [1] | tap water | 96.8–104.0 | $2.2 \times 10^{-10}$ | [52] |
| | Intercalated | Tetramethylrhodamine | Fluorescence | Truncated BPA-Apt-1 [2] | tap water | 95–104 | $5 \times 10^{-7}$ | [77] |
| | External | Berberine | Fluorescence | Modified BPA-Apt-2 [3] Bisphenol A-aptamer$_1$ [4] | tap water | 92.4–102.3 | $3.2 \times 10^{-8}$ | [79] |
| | Incorporated into DNA sequence | Carboxyfluorescein/Dabcyl | Fluorescence | BPA-Apt-2 | milk | 96–106.4 | $5.0 \times 10^{-14}$ | [78] |
| | External | Cadmium Tellured Quantum Dots (CdTe QDs) | Fluorescence | BPA-Apt-2 | tap water | 95.5–102 | $8.15 \times 10^{-12}$ | [75] |
| | - | 4-cyano-4′-pentylbiphenyl | Polarized light microscopy | BPA-Apt-1 | orange juice | not reported | $6.0 \times 10^{-10}$ | [62] |
| | External | Gold nano particles (AuNPs) | Color | BPA-Apt-2 | tap water | 100.9–112.7 | $1.5 \times 10^{-9}$ | [76] |

**Table 2.** *Cont.*

| Type of Aptasensor | Probe | | Detection Technique | Aptamer Sequence | Food Tested for Practical Application | Recovery (%) | LOD (M) | References |
| | Method of Use | Type | | | | | | |
|---|---|---|---|---|---|---|---|---|
| | External | Gold nano particles (AuNPs) | Color | BPA-Apt-3 | rice | not reported | $4.4 \times 10^{-12}$ | [67] |
| | Conjugated | AuNPs | Color | BPA-Apt-4 BPA-Apt-5 | mineral water | 98.33–102.78 | $7.60 \times 10^{-15}$ BPA-Apt-4 $1.441 \times 10^{-14}$ BPA-Apt-5 | [46] |
| | | | | | milk | 96–97.08 | | |
| | | | | | orange juice | 98.26–02.83 | | |
| RT-qPCR | - | - | Cycle threshold (Ct) values | BPA-Apt-2 | tap water | 96.0–104.5 | $7.0 \times 10^{-10}$ | [57] |

[1] The aptamer used in the biosensor contains an additional tail (GGCTACGAGGGAAATGCGGT) at 5′-end. [2] Residues from 36 to 60 of the original BPA-apt-1 has been truncated. [3] The alignment of the sequence used by the authors (named aptamer-2) and that of BPA-Apt-2 leads to cutting of the residues $G_{10}G_{11}T_{12}$ from the latter. The resulting aptamer was 60 nts. [4] We tagged Bisphenol A-aptamer$_1$ with the sequence (5′-TGG TCG TTG GTC GTT CGC GTT TCT GGA TTT TTT ATT TCT GGG GTT CAG TTC TTT TTT GT-3′) as the authors have done. No further information about this aptameric sequence has been reported.

## 5. Conclusions and Future Perspectives

Several studies have been successfully carried out to obtain aptasensors with high sensitivity. In December 2021 the EFSA proposed to considerably lower the TDI for BPA. This can lead to a change of BPA SMLs in the future. In this framework, aptasensors can represent a promising, useful, and reliable tool to monitor BPA levels in food samples. However, some improvements are required.

First of all, aptasensors have been tested on a limited number of food types. Further efforts are needed to evaluate the performance of the same or new BPA aptasensors for a broader selection of food categories and for more complex matrices than liquid foods. Regarding the latter, it should be pointed out that the binding of the aptamer to a target molecule depends on its folding in a unique structure, which, in turn, is strongly influenced by the environment where the analyte recognition occurs. In complex food matrices, molecules other than the target one can affect the aptamer/BPA complex formation and consequently the reproducibility of the results. Therefore, the need for pre-treatment of the sample cannot be ruled out *a priori*.

Secondly, all studies lack a complete validation process in accordance with the specific guidelines for the validation of analytical methods in food safety studies. Then, the concrete applicability of aptasensors in food analyses for BPA risk assessment needs to be verified.

Finally, it is important to note that BPA analogs are now used to replace BPA in FCMs, although several studies have highlighted that they have a toxicity comparable to that of BPA. As a result, the monitoring of their levels is also mandatory to protect consumers' health. For this reason, another challenge of the future is the development of new sensitive aptasensors that are able to rapidly, simply, and sensitively quantify both BPA and its substitutes in a multi-analyte simultaneous analysis.

**Author Contributions:** Writing—review and editing, M.E.S., A.A., M.V. and S.A. All authors have read and agreed to the published version of the manuscript.

**Funding:** This research received no external funding.

**Institutional Review Board Statement:** Not applicable.

**Informed Consent Statement:** Not applicable.

**Data Availability Statement:** Not applicable.

**Conflicts of Interest:** The authors declare no conflict of interest.

## Abbreviations

ACET: Alternating Current Electro-Thermal Effect; Au-PtNPs, Golden-Platinum NanoParticles; AgCNFs, Silver nanoparticles/Carbon NanoFibres composite; AgNPs, Silver Nano-Particles; ACF, ACriFlavine; Apt, Aptamer; AuNPs, Gold Nano Particles; BDD, Boron-Doped Diamond; BPA, BisPhenol A; BPB, BisPhenol B; BPS, BPF, BisPhenol F; BisPhenol S; CdTe QDs, Cadmium Tellured Quantum Dots; CNTs-COOH, Carbon Nanotubes-COOH; Cy3, Cyanine dye 3; dsDNA, double-stranded DNA; cDNA, capture DNA; hDNA, help DNA; mDNA, message DNA; rDNA, report DNA; DNAzyme, deoxyribozymes; EDC, Endocrine-Disrupting Chemical; EFSA, European Food Safety Authority; Exo III, Exonuclease III; ECL, ElectroChemiLuminescence; FCM, Food Contact Material; GCE, Glassy Carbon Electrode; GO, Graphene Oxide; HCR, Hybridization Chain Reaction; HQ, HydroQuinone; HRP, HorseRadish Peroxidase; IFE, Inner Filter Effect; ITO, Indium-Tin-Oxide; LC, Liquid Crystal; f-MWCNTs, functionalized Multi-Wall Carbon NanoTubes; PDDA, Poly(DiallylDimethylAmmonium chloride); PEC, Photo-ElectroChemical; PPY, PoliPYrrole; RT-qPCR, Real-Time quantitative Polymerase Chain Reaction; SERS, Surface-Enhanced Raman Scattering; SML, Specific Migration Limit; SPGE, Screen-Printed Gold Electrode; TDI, Tolerable Daily Intake; TdT, Terminal deoxynucleotidyl Transferase; dT, 2′-deoxyTimine; TMR, TetraMethylRhodamine; UCNPs, Upconversion NanoPparticles; WE, Working Electrode.

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
