# Peer review of "Aptamer-Based Biosensors for the Analytical Determination of Bisphenol A in Foodstuffs"

_applsci, doi:10.3390/app12083752_

Round 1

Reviewer 1 Report

Dear writers,

                    According to my opinion this review is very good and descriptive of the field that you want to presented. The only thing that I would like to see is the table that describe the abbreviations. For Example, BPA= Bisphenol A, etc.

For this reason, I put it in accept minor revision.

Kind regards

Author Response

Referee 1

…….. The only thing that I would like to see is the table that describe the abbreviations. For Example, BPA= Bisphenol A, etc.

In accordance with the referee’s comment the authors have added Abbreviation paragraph after Keywords.

Reviewer 2 Report

This paper reviews research on aptasensor designed to detect BPA in food matrices. This review paper focuses on the progresses in the field of BPA aptamer strategies, filling a gap in the field of reviewing aptasensor. In my opinion, this review paper could be accepted for publication on the journal Applied science after the revisions as follows.

*The keywords should be separated by a semicolon. Omit the keywords that are identical to terms contained in the title. Search engines will find such terms in the title already because the title is searched first (before the keywords). Note that an article will be found more often by search engines if title, abstract and keywords well reflect the contents of the article.

*The challenges and future perspective are missing and is suggested to contain more information about future perspective.

*Please include the distinct advantages of aptasensor. Additionally, please discuss the possible strengths and limitations of aptasensor in the field of sensing.

* Abstract doesn’t present the clear picture of review paper. Explain briefly about your objective in last paragraph.

*Check reference carefully. It contains too many errors regarding format such as font size, case. etc.  Additionally, the reference citations are also wrong such as line#72, 77, 94,105, 109, 208, 349 (ref. [20], [21-29], [34] [35] [36], [32], [70]). Go through the whole manuscript to remove ambiguity and flaws regarding citation and references style and format.

*The level of English throughout the manuscript is very poor and manuscript contain grammatical and format errors such as Line#65, The same Regulation (used lower case for regulation). Check Copyright permission caption in figure 2 and 3 related to font size and text annotation. Table 2, formatting was ignored (uppercase and lowercase) such as AptaSensor should be Aptasensor, electrochemical. There are formatting and spacing mistakes.

* The sensing mechanisms of aptasensor are not well-organized in this review. Give some brief overview about aptasensor.

* There is also need to provide the graphical description.

* The manuscript lacks of personal opinions.

Author Response

Referee 2

1 *The keywords should be separated by a semicolon. Omit the keywords that are identical to terms contained in the title. Search engines will find such terms in the title already because the title is searched first (before the keywords). Note that an article will be found more often by search engines if title, abstract and keywords well reflect the contents of the article.

- We have very appreciated this referee’s comment. In accordance with it, we have substituted the previous keywords BPA and aptamers with bisphenols and Endocrine Disrupting Chemicals, respectively.

2*The challenges and future perspective are missing and is suggested to contain more information about future perspective.

- We have re-organized the conclusion paragraph including our considerations and feature perspectives. We have also added featured application paragraph before Abstract according to the Applied Sciences template.

3*Please include the distinct advantages of aptasensor. Additionally, please discuss the possible strengths and limitations of aptasensor in the field of sensing.

In accordance with the reviewer’s comments, we have introduced, along with the section 4 the distinct advantages of the discussed aptasensors. Some considerations about this point have been also introduced in the sections 1 and 5, introduction and conclusion respectively, by illustrating advantages and limitations of aptasensors in food analysis.

4* Abstract doesn’t present the clear picture of review paper. Explain briefly about your objective in last paragraph.

- We have rearranged the Abstract to fulfill the referee’s request and the word number limit for the Abstract, as indicated in submission guidelines.

5*Check reference carefully. It contains too many errors regarding format such as font size, case. etc.  Additionally, the reference citations are also wrong such as line#72, 77, 94,105, 109, 208, 349 (ref. [20], [21-29], [34] [35] [36], [32], [70]). Go through the whole manuscript to remove ambiguity and flaws regarding citation and references style and format.

Thank you for this observation. We have checked all references for formatting mistakes.

6*The level of English throughout the manuscript is very poor and manuscript contain grammatical and format errors such as Line#65, The same Regulation (used lower case for regulation). Check Copyright permission caption in figure 2 and 3 related. Table 2, formatting was ignored (uppercase and lowercase) such as AptaSensor should be Aptasensor, electrochemical. There are formatting and spacing mistakes.

We apologize for mistakes and are very grateful to the reviewer for showing it to our attention. We have checked the level of English, font size and text annotation of figures 2 and 3 and formatting of table 2.

7* The sensing mechanisms of aptasensor are not well-organized in this review. Give some brief overview about aptasensor.

8* There is also need to provide the graphical description.

In accordance with this referee’s comments, we have revised section 4 by modifying its structure and contents and by adding two new figures, 5 and 6.

 9* The manuscript lacks personal opinions.

In accordance with this referee’s comment, we have re-organized the sections 1, 4, and 5 (introduction, Detection techniques applied in the analysis of food real samples and conclusion) by adding up our point of view on advantages and limits of aptasensors and feature perspectives.

Reviewer 3 Report

I recommend expanding the “thesis statement” at the end of the introductory section to better motivate the subject of the review. Why is this an interesting topic to write about now—for instance, have there been significant recent developments in how this topic is understood or practiced, does this article provide a new interpretation of an existing field, or is it simply the first comprehensive review of a timely subject? Finally, what will a reader learn from this article that he or she couldn’t learn simply by reading the references?

Major:

Aptamer’s incoherencies: Aptamers are single stranded oligonucleotides not double stranded, DNA/ RNA are already oligonucleotides (Pag 3 line 100). Aptamers are selected not developed by SELEX (Pag 3 line 104). SELEX can be performed using DNA or RNA (Pag 3 Line 106). Figure 2 represents the secondary structure and not the three-dimensional conformation of aptamers.

Table 2: Add the Kd of each aptamer. It is a very important property to evaluate aptamers.

Pag 6 Line 205: Rephrase the paragraph. The sentences are contradictory.

Minor:

Pag 1 line 36: is present instead of occurs

Pag 1 line 37: The second one instead of additionally

Pag 3 line 110: Remove both

Pag 3 line 112: Rephrase the sentence.

Author Response

Referee 3

I recommend expanding the “thesis statement” at the end of the introductory section to better motivate the subject of the review. Why is this an interesting topic to write about now—for instance, have there been significant recent developments in how this topic is understood or practiced, does this article provide a new interpretation of an existing field, or is it simply the first comprehensive review of a timely subject? Finally, what will a reader learn from this article that he or she couldn’t learn simply by reading the references?

In accordance with this reviewer’ s comment we have expanded the thesis statement at the end of  the introduction.

Major:

Aptamer’s incoherencies: Aptamers are single stranded oligonucleotides not double stranded, DNA/ RNA are already oligonucleotides (Pag 3 line 100). Aptamers are selected not developed by SELEX (Pag 3 line 104). SELEX can be performed using DNA or RNA (Pag 3 Line 106). Figure 2 represents the secondary structure and not the three-dimensional conformation of aptamers.

We have accepted referee comments and we have modified the text in agreement with them.

Table 2: Add the Kd of each aptamer. It is a very important property to evaluate aptamers.

We searched for Kd values as suggested by the referee. We have added the available Kd values in table 1.

Pag 6 Line 205: Rephrase the paragraph. The sentences are contradictory.

In accordance with the referee’s suggestion, we have modified the corresponding paragraph.

Minor:

Pag 1 line 36: is present instead of occurs

Pag 1 line 37: The second one instead of additionally

Pag 3 line 110: Remove both

Pag 3 line 112: Rephrase the sentence.

We have modified the text in agreement with the referee’s suggestions.

Round 2

Reviewer 2 Report

The review was greatly improved and accepted

Reviewer 3 Report

The manuscript has been sufficiently improved to warrant publication in Applied Sciences.